

# Impact of paternal hepatitis B on pregnancy outcomes in couples undergoing assisted reproductive technology treatment: a systematic review and meta-analysis

Juanting Gao[1], Qiyin Dong[2], Liping Shen[1] and Xiuping Zhu[3]

[1] Department of Emergency, Huzhou Maternity & Child Health Care Hospital, Huzhou, Zhejiang, China
[2] Department of Reproductive Center, Huzhou Maternity & Child Health Care Hospital, Huzhou, Zhejiang, China
[3] Department of Outpatient, Huzhou Maternity & Child Health Care Hospital, Huzhou, Zhejiang, China

Corresponding author
Xiuping Zhu, maydayzxp@163.com

## ABSTRACT

**Background:** This study intends to evaluate the influence of hepatitis B virus (HBV) infection on clinical pregnancy rate (CPR) and live birth rate (LBR) per woman and cycle in couples who achieved pregnancy through assisted reproductive technology (ART).

**Methods:** PubMed, Embase, CNKI, Scopus, Web of Science, and Wangfang databases were comprehensively searched for articles reporting data on pregnancies achieved through ATR and providing information on the paternal HBV status and CPR and LBR. A random-effects model was used for the meta-analyses to pool odds ratios (OR) with corresponding confidence intervals (CI). Subgroup analyses were done based on the matching status.

**Results:** The analysis included 10 studies (4,848 participants) for CPR per woman, 10 studies (28,951 participants) for CPR per cycle, four studies (2,327 participants) for LBR per woman, and seven studies (26,324 participants) for LBR per cycle. The analysis showed no considerable association between the HBV status and the CPR or LBR, with the pooled OR of 1.015 (95% CI [0.860–1.199], $p = 0.857$) for CPR per woman and of 1.051 (95% CI [0.870–1.271], $p = 0.603$) for CPR per cycle. Pooled OR for LBR per woman was 0.852 (95% CI [0.717–1.012], $p = 0.068$), and for LBR per cycle was 0.999 (95% CI [0.851–1.172], $p = 0.987$).

**Conclusions:** Paternal HBV infection does not significantly affect clinical pregnancy or live birth rates in ART pregnancies. Our findings inform clinical practice and provide reassurance to couples undergoing ART that paternal HBV infection should not be a major concern in terms of pregnancy outcomes.

## INTRODUCTION

Chronic hepatitis B virus (HBV) infection affects approximately 257 million people worldwide (*Tan et al., 2021*) and remains a leading cause of cirrhosis and hepatocellular carcinoma (*Tripathi & Mousa, 2024*). While maternal HBV transmission and its perinatal consequences—such as preterm birth and neonatal infection—have been extensively documented (*Umar et al., 2013*; *Nelson, Jamieson & Murphy, 2014*), the potential impact of paternal HBV infection on fertility and assisted reproductive technology (ART) outcomes has received comparatively little attention. ART techniques, including *in vitro* fertilization and intracytoplasmic sperm injection, have revolutionized infertility treatment (*Jain & Singh, 2024*; *National Institute for Health and Care, 2017*), yet current guidelines focus almost exclusively on maternal screening and management, leaving a critical gap in our understanding of paternal contributions.

A smaller number of studies have examined the role of paternal HBV in ART, reporting possible associations with miscarriage, preterm delivery, and low birth weight (*Cito et al., 2021*; *Sirilert & Tongsong, 2021*; *Zhu et al., 2024*). However, heterogeneity in study design, population characteristics, and ART protocols has precluded definitive conclusions (*Jia et al., 2022*). Biological plausibility is supported by evidence that HBV can persist in semen, potentially affecting sperm function, embryogenesis, or the early immunological environment at implantation (*Navabakhsh et al., 2011*; *Kanji et al., 2019*; *Nguyen et al., 2020*). Proposed interventions—such as refined sperm-washing techniques and preconception antiviral therapy for HBV-positive men—underscore the importance of robust, evidence-based guidance (*Jaffe & Brown, 2017*).

Against the backdrop of rising ART utilization and a substantial global HBV burden, a systematic evaluation of paternal HBV infection's impact on reproductive outcomes is both timely and necessary. This review synthesizes all available data on clinical pregnancy rate (CPR) and live birth rate (LBR) in couples undergoing ART, critically appraises study quality, and explores sources of heterogeneity (*e.g.*, matching status). The primary research question is: Does paternal HBV infection affect clinical pregnancy rates and live birth rates in couples undergoing ART?

## METHODS

### Study design and protocol registration

This review followed the guidelines outlined in the PRISMA statement (*Page et al., 2021*), and was preregistered in the International Prospective Register of Systematic Reviews (PROSPERO) with the registration number (CRD42024559724).

### Eligibility criteria

*Population*: Couples undergoing ART, such as in-vitro fertilization (IVF) and intracytoplasmic sperm injection (ICSI).

*Exposure*: Paternal HBV infection, defined by seropositivity to HBV surface- and/or hepatitis B e-antigens.

*Comparators*: Couples with HBV-negative paternal partners or with maternal HBV infection only.

*Outcomes*: Primary outcomes of this review as follows:

*Clinical pregnancy rate (CPR) per woman*: proportion of women who achieved at least one intrauterine gestational sac on transvaginal ultrasound out of the total number of women who commenced an ART treatment cycle.

*CPR per cycle*: number of treatment cycles that resulted in a clinical pregnancy divided by the total number of ART cycles initiated.

*Live birth rate (LBR) per woman*: proportion of women who delivered at least one live infant (beyond 24 weeks' gestation) out of the total women treated.

*LBR per cycle*: number of ART cycles resulting in a live birth divided by all cycles started.

*Study design*: Cohort, case-control, or cross-sectional studies.

*Reporting*: Studies that reported sufficient data to calculate relative risks (RR), odds ratios (OR), or mean differences with 95% confidence intervals (CI).

## Literature searches

PubMed (Medline), Embase, Web of Science, Scopus, China National Knowledge Infrastructure (CNKI), and Wangfang databases were searched from inception to June 2024 using a combination of Medical Subject Headings (MeSH) terms and free-text keywords related to HBV, ART, and pregnancy outcomes. Additionally, bibliography sections and appropriate reviews were manually screened for relevant studies. The detailed search strategy in each of the databases are provided in Supplemental File.

## Study selection

All records retrieved from the six databases were first imported into EndNote (version X9), where duplicates were automatically identified and removed. Two reviewers (JG and QD) then independently screened the titles and abstracts of the remaining articles against our predefined eligibility criteria. Articles deemed potentially relevant were retrieved in full text and assessed in detail by both reviewers. At each stage (title/abstract and full-text), any disagreement was resolved by discussion, with referral to a third reviewer (LS) when consensus could not be reached. Reasons for exclusion at the full-text stage (*e.g.*, wrong population, lack of pertinent outcomes, or insufficient data) were documented. The entire process is summarised in the PRISMA flow diagram (Fig. 1), which details the numbers of records identified, screened, excluded (with primary reasons), and finally included in the synthesis.

## Data extraction

A standardized extraction form was used by the two authors (JG and QD) to independently extract study characteristics (first author, year of publication, country, study
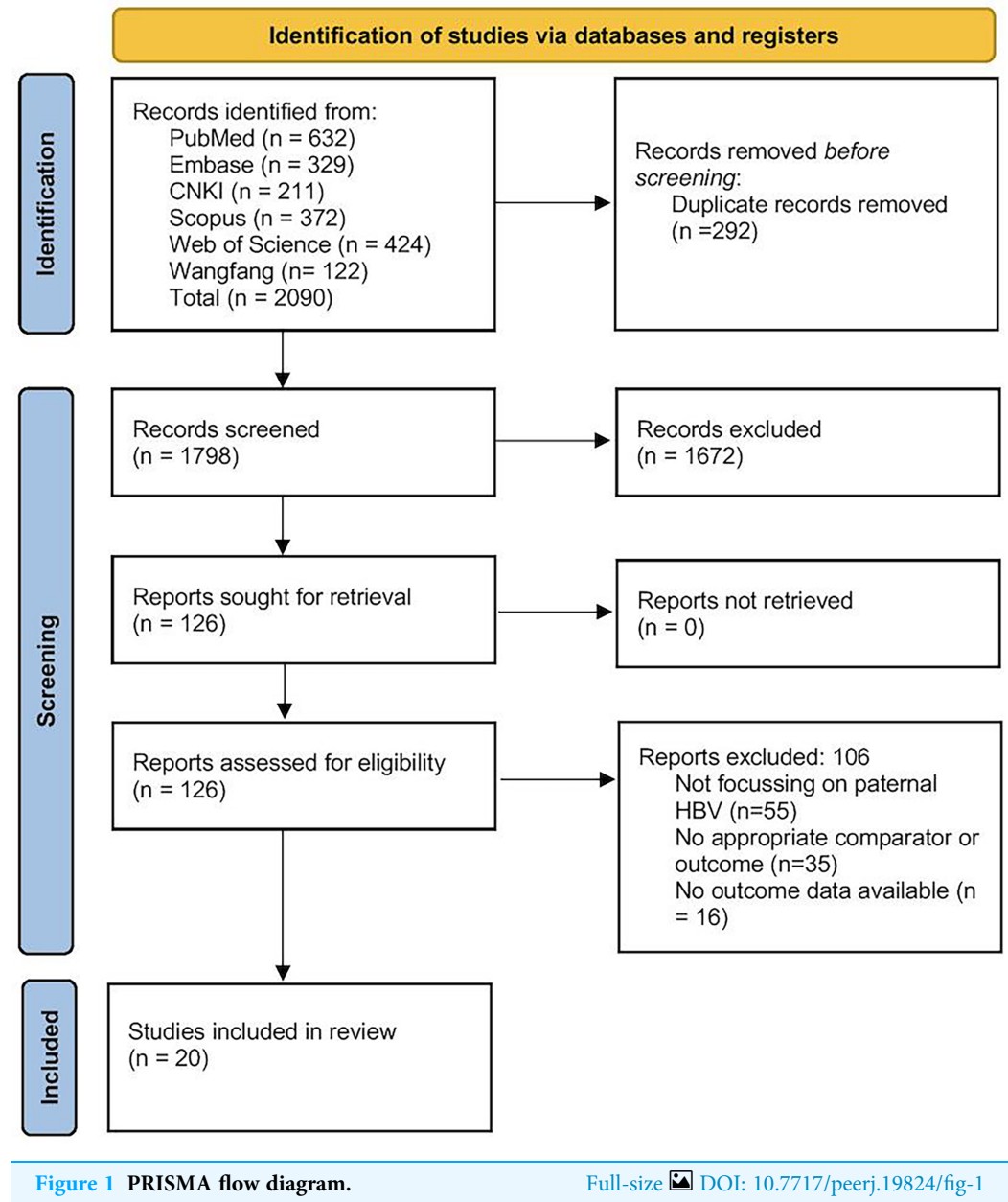

**Figure 1 PRISMA flow diagram.**

design, sample size, and duration), population characteristics (age, type of ART procedure, and HBV infection status), outcomes (clinical pregnancy and LBR), and methodological quality.

## Risk of bias assessment

The Newcastle-Ottawa Scale (NOS) was employed to assess the risk of bias based on the selection of study groups, comparability of groups, and ascertainment of exposure/outcome. A score of 7–9 points was considered lower risk of bias, 4–6 points moderate risk of bias, and 0–3 points as high risk of bias (*Wells et al., 2015*).

## Statistical analysis

Meta-analyses were done using a random-effects model. RR and OR were calculated for binary outcomes, and mean differences were calculated for continuous outcomes, with their 95% confidence intervals (CI). The $I^2$ statistic measured heterogeneity, with $I^2$ of 25% as low, $I^2$ of 50% as moderate, and $I^2$ of 75% as high heterogeneity (*Cumpston et al., 2019*).

Studies were classified as matched if the investigators explicitly paired or frequency-matched HBV-positive and HBV-negative groups on one or more key baseline characteristics (*e.g.*, age, sex, cirrhosis status), or if they conducted a multivariable or propensity-score analysis that controlled for those same baseline covariates (*e.g.*, adjusted odds ratio or hazard ratio in regression models). Studies that compared the two groups without such matching at the design stage or without covariate adjustment on the same baseline variables were considered unmatched. Subgroup analyses by matching status were then performed to explore whether balancing baseline covariates influenced the pooled effect estimates.

Funnel plots and Egger's test assessed publication bias, with $p < 0.05$ being significant. Results were synthesized narratively and quantitatively, with detailed tables (for individual study characteristics) and forest plots illustrating the main findings (*Cumpston et al., 2019*). Sensitivity analysis was performed by excluding studies with women HBV infection and high risk of bias and reported the revised pooled estimate, which in turn helps to check the robustness of the estimates. STATA software (version 17.0) was used for the analysis.

## RESULTS

### Search results

Of the initial pool of 1,798 records identified by the literature search, 126 reports were assessed for eligibility, and 106 were eliminated. The reasons for exclusion were as follows: studies not focusing on paternal HBV, lack of appropriate comparator or outcome, or missing outcome data (Fig. 1). Eventually, 20 articles were included in the review (*Zhao et al., 2007*; *Lam et al., 2010*; *Lee et al., 2010*; *Zhou et al., 2011*; *Oger et al., 2011*; *Chen et al., 2013*; *Bu et al., 2014*; *Du et al., 2014*; *Ye et al., 2014*; *Shi et al., 2014*; *Zheng et al., 2016*; *Kong et al., 2017*; *Lubis et al., 2018*; *He et al., 2018*; *Wang et al., 2019*; *Cito et al., 2021*; *Ma et al., 2023*; *Meng et al., 2024*; *Sun et al., 2024*; *Zhu et al., 2024*).

### Characteristics of the included studies

Characteristics of the included papers are provided in Table 1. Most (16 out of 20 studies) were retrospective cohort studies, while rest of them were case control studies. Most (17 out of 20 studies) were done in China, while Italy, Indonesia and France provides other three studies. Among retrospective cohorts, we designated a study as 'matched' only if the authors explicitly reported either (1) design-level pairing/frequency matching on baseline covariates or (2) multivariable adjustment or propensity-score matching using those same covariates. Case–control studies that did not clearly report matching or adjusted analyses on the key confounders (age, sex, cirrhosis) were considered 'unmatched'. Sample sizes in the HBV-positive exposed groups ranged widely from 9 to 1,528, with control groups generally two- to tenfold larger. ART modalities included IVF alone, ICSI alone, combined

**Table 1 Summary of the characteristics of the contributing research.**

| Study identifier | Study region | Design | Participant details | Type of assisted reproductive technology | Sample size (paternal HBV-positive group) | Sample size in the control group | Maternal Mean age (in years) | Paternal mean age (in years) | Matching done or not |
|---|---|---|---|---|---|---|---|---|---|
| Bu et al. (2014) | China | Retrospective cohort study | Sub fertile patients undergoing oocyte donation | IVF | 20 | 257 | I = 35 ± 2.3<br>C = 36.3 ± 2.4 | I = 35 ± 1.7<br>C = 36 ± 2.5 | Unmatched |
| Chen et al. (2013) | China | Retrospective study | Couples with men HBV infection | IVF/ICSI | 115 | 230 | I = 31.3 ± 4.3<br>C = 31.4 ± 4.1 | NA | Matched |
| Cito et al. (2021) | Italy | Retrospective cohort study | Infertile couples referred to ARTs Centre for the first ART treatment | IVF/ICSI | 66 | 68 | I = 34.0 ± 1.7<br>C = 36 ± 1.3 | I = 37.0 ± 2.2<br>C = 38 ± 1.5 | Matched |
| Du et al. (2014) | China | Retrospective study | HBV infected and uninfected couple | IVF/ICSI | 637 | 1,892 | NA | NA | Unmatched |
| He et al. (2018) | China | Retrospective cohort study | Couples with an HBsAg-positive male partner and an HBsAg-negative female partner | IVF/ICSI | 39 | 78 | I = 31 ± 6.5<br>C = 31.5 ± 3.8 | NA | Unmatched |
| Kong et al. (2017) | China | Retrospective cohort study | Fathers of IVF embryo with chronic HBV infection | IVF | 9 | 50 | NA | I = 29.9 ± 5.1<br>C = 30.4 ± 5.5 | Unmatched |
| Lam et al. (2010) | China | Retrospective cohort study | Couples undergoing IVF and embryo transfer cycles | IVF—ET | 56 | 231 | NA | I = 35.2 ± 0.4<br>C = 35.7 ± 0.3 | Matched |
| Lee et al. (2010) | China | Retrospective study | Subfertile couples undergoing their first IVF cycle | IVF/ICSI | 154 | 1,473 | I = 35 ± 0.8<br>C = 35 ± 0.7 | NA | Unmatched |
| Lubis et al. (2018) | Indonesia | Retrospective analytic study | HBV-infected and non-HBV infected male partner groups who have been treated with *in vitro* fertilization (IVF) | IVF | 17 | 84 | I = 35.4 ± 3.4<br>C = 32.9 ± 4.7 | I = 37.6 ± 4.0<br>C = 34.7 ± 5.6 | Unmatched |
| Ma et al. (2023) | China | Retrospective study | Infertility patients undergoing their first embryo transfer cycles by ART treatment | IVF/ICSI | 1,528 | 19,561 | I = 29.65 ± 3.81<br>C = 30.25 ± 4.08 | NA | Unmatched |
| Meng et al. (2024) | China | Retrospective cohort study | Infertile couples aged 24 to 40 years, who were admitted to a reproductive medicine center to undergo ART for the first time | IVF/ICSI | 469 | 1,112 | I = 30 ± 2.5<br>C = 31 ± 2.1 | I = 31 ± 2.5<br>C = 32 ± 1.5 | Unmatched |
| Oger et al. (2011) | France | Case control study | Couples in which male partners were hepatitis B positive | IVF | 32 | 64 | I = 32 ± 5.2<br>C = 32 ± 5.0 | I = 34.7 ± 5.0<br>C = 35.5 ± 6.3 | Matched |
| Shi et al. (2014) | China | Case control study | At least one partner being HBsAg-seropositive undergoing their first IVF and embryo transfer cycle | IVF-ET | 213 | 426 | I = 30.5 ± 3.9<br>C = 30.4 ± 4.5 | I = 32.9 ± 4.7<br>C = 32.6 ± 5.0 | Matched |

| Study identifier | Study region | Design | Participant details | Type of assisted reproductive technology | Sample size (paternal HBV-positive group) | Sample size in the control group | Maternal Mean age (in years) | Paternal mean age (in years) | Matching done or not |
|---|---|---|---|---|---|---|---|---|---|
| *Sun et al. (2024)* | China | Retrospective study | Infertile couples undergoing intrauterine insemination treatment | IUI | 212 | 750 | I = 30 ± 1<br>C = 29 ± 1 | I = 32 ± 1<br>C = 31 ± 1 | Unmatched |
| *Wang et al. (2021)* | China | Retrospective cohort study | Infertile couples with male HBV infection undergoing embryo transfer and for IUI | IUI/IVF/ICSI | 227 | 454 | I = 33.0 ± 4.0<br>C = 33.0 ± 3.9 | I = 34.8 ± 4.9<br>C = 34.7 ± 4.7 | Matched |
| *Ye et al. (2014)* | China | Retrospective study | Couples who received ART treatment | IVF/ICSI | 25 | 50 | I = 29.7 ± 5.9<br>C = 29.2 ± 5.7 | I = 30.6 ± 4.9<br>C = 30.8 ± 5.1 | Unmatched |
| *Zhao et al. (2007)* | China | Case control study | Couples receiving IVF-ET with the husbands (but not the wives) positive for hepatitis B surface antigen | IVF-ET | 102 | 204 | NA | NA | Unmatched |
| *Zheng et al. (2016)* | China | Retrospective study | Female partner tested negative for HBV DNA, HBsAg, HBeAg, HBeAb and HBcAb among couples whom undergo first Intra cytoplasmic sperm injection | ICSI | 217 | 121 | I = 30.7 ± 4.1<br>C = 30.6 ± 3.8 | I = 32.4 ± 5.2<br>C = 32.7 ± 5.1 | Unmatched |
| *Zhou et al. (2011)* | China | Case control study | Male patients seeking fertility assistance | IVF/ICSI–ET | 457 | 459 | I = 31.4 ± 4.9<br>C = 31.4 ± 4.9 | I = 33.4 ± 4.8<br>C = 33.5 ± 4.8 | Matched |
| *Zhu et al. (2024)* | China | Retrospective cohort study | Couples who received fresh embryo transfer cycles for the first time | IVF/ICSI | 821 | 821 | I = 33 ± 1<br>C = 33 ± 1.2 | I = 32 ± 1<br>C = 31 ± 1 | Matched |

**Note:**
NA, Not available; IUI, intra uterine insemination; IVF, *in vitro* fertilization; ICSI, intra cytoplasmic sperm injection; ET, embryo transfer.

IVF/ICSI, IVF–ET, and IUI or mixed IUI/IVF/ICSI. Mean parental ages were comparable between groups (maternal 29–35 years; paternal 29–38 years), and matching on age or key covariates was explicitly performed in eight studies, while rest of the studies were unmatched. Table 2 shows the risk of bias assessment of the included studies. Half of the included studies (10 out of 20 studies) had lower risk of bias, while seven studies had moderate risk of bias and only three studies had higher risk of bias.

## Impact of paternal HBV infection on clinical pregnancy rate (per woman)

The meta-analysis investigating the impact of paternal HBV infection on CPR per woman included 10 studies encompassing 4,848 participants. The pooled OR for CPR among women whose partners were HBV-positive was 1.015, with 95% CI ranging from [0.860 to 1.199] (Fig. 2). This indicates that the CPR in women with or without HBV-positive partners was comparable ($p = 0.857$). Heterogeneity was lower ($I^2 = 25.1\%$; $p = 0.213$).

**Table 2  Risk of bias assessment.**

| Study identifier | Selection domain | Comparability domain | Outcome domain | Risk of bias |
|---|---|---|---|---|
| Bu et al. (2014) | 3 points | 2 points | 3 points | Low (8 points) |
| Chen et al. (2013) | 3 points | 2 points | 2 points | Low (7 points) |
| Cito et al. (2021) | 2 points | 2 points | 2 points | Moderate (6 points) |
| Du et al. (2014) | 2 points | 1 point | 2 points | Moderate (5 points) |
| He et al. (2018) | 3 points | 2 points | 2 points | Low (7 points) |
| Kong et al. (2017) | 3 points | 2 points | 3 points | Low (8 points) |
| Lam et al. (2010) | 2 points | 2 points | 1 point | Moderate (5 points) |
| Lee et al. (2010) | 1 point | 1 point | 1 point | High (3 points) |
| Lubis et al. (2018) | 3 points | 2 points | 2 points | Low (7 points) |
| Ma et al. (2023) | 0 point | 1 point | 1 point | High (2 points) |
| Meng et al. (2024) | 3 points | 2 points | 2 points | Low (7 points) |
| Oger et al. (2011) | 2 points | 2 points | 3 points | Low (7 points) |
| Shi et al. (2014) | 2 points | 1 point | 2 points | Moderate (5 points) |
| Sun et al. (2024) | 2 points | 2 points | 2 points | Moderate (6 points) |
| Wang et al. (2021) | 2 points | 2 points | 3 points | Low (7 points) |
| Ye et al. (2014) | 3 points | 2 points | 3 points | Low (8 points) |
| Zhao et al. (2007) | 2 points | 1 point | 1 point | Moderate (4 points) |
| Zheng et al. (2016) | 2 points | 2 points | 2 points | Moderate (6 points) |
| Zhou et al. (2011) | 3 points | 2 points | 2 points | Low (7 points) |
| Zhu et al. (2024) | 1 point | 1 point | 1 point | High (3 points) |

Sensitivity analysis by excluding studies with higher risk of bias showed that the pooled OR was 1.035 (95% CI [0.830–1.290]), indicating the robustness of the overall pooled effect size.

Subgroup analysis revealed that in unmatched studies, pooled OR was 0.925 (95% CI [0.670–1.277], $p = 0.635$), while in matched studies, pooled OR was 1.009 (95% CI [0.855–1.191], $p = 0.913$). Heterogeneity was moderate in unmatched studies ($I^2 = 41.6\%$; $p = 0.114$) and low in matched studies ($I^2 = 0.0\%$; $p = 0.420$). Overall, CPR had no significant association with paternal HBV positivity in both subgroups (Fig. 3), with a symmetrical funnel plot (Fig. S1) and a non-significant Egger's test ($p = 0.361$).

**Impact of paternal HBV infection on clinical pregnancy rate (per cycle)**

The meta-analysis investigating the effect of paternal HBV infection on CPR per cycle included 10 studies with 28,951 participants. Analysis of the CPR for cycles involving HBV-positive men indicated a non-significant association (OR of 1.051, with a 95% CI of [0.870 to 1.271]) ($p = 0.603$) (Fig. 4) with moderate ($I^2$ value of 64.6%; $p = 0.003$) heterogeneity. Sensitivity analysis by excluding studies with higher risk of bias showed that the pooled OR was 1.059 (95% CI [0.790–1.420]), indicating the robustness of the overall pooled effect size.

Subgroup analysis based on matching revealed that in unmatched studies, pooled OR for CPR was 0.985 (95% CI [0.833–1.164], $p = 0.857$), while in matched studies, pooled OR

**Figure 2 Forest plot showing impact of paternal HBV infection on clinical pregnancy rate (per women)** (*Bu et al., 2014*; *Kong et al., 2017*; *Lubis et al., 2018*; *Meng et al., 2024*; *Shi et al., 2014*; *Wang et al., 2021*; *Ye et al., 2014*; *Zhao et al., 2007*; *Zheng et al., 2016*; *Zhu et al., 2024*).

was 1.187 (95% CI [0.717–1.968], $p = 0.505$). Heterogeneity was moderate in unmatched studies ($I^2 = 47.7\%$; $p = 0.105$) and high in matched studies ($I^2 = 73.2\%$; $p = 0.005$). Overall, there was no significant link between paternal HBV-positivity and the CPR per cycle in both subgroups (Fig. 5). The pooled adjusted RR calculated based on the adjusted effect size was 0.733 (95% CI [0.453–1.014]) (Fig. 6). The symmetrical funnel plot (Fig. S2) and non-significant Egger's test ($p = 0.816$) confirmed a lack of publication bias.

## Impact of paternal HBV infection on live birth rate (per woman)

The meta-analysis investigating the effect of paternal HBV infection on LBR per woman included four studies with 2,327 participants. Pooled OR for LBR among women whose partners were HBV-positive was 0.852, with 95% CI ranging from [0.717 to 1.012] ($p = 0.068$), indicating comparable LBR in all pregnancies regardless of the paternal HBV status (Fig. 7) with no heterogeneity ($I^2 = 0.0\%$). Sensitivity analysis by excluding studies with higher risk of bias showed that the pooled OR was 1.021 (95% CI [0.725–1.437]), indicating the robustness of the overall pooled effect size. Subgroup analysis based on matching status was not done as there was only one study in one of the subgroups. Publication bias assessment was also not done due to the small number of studies.

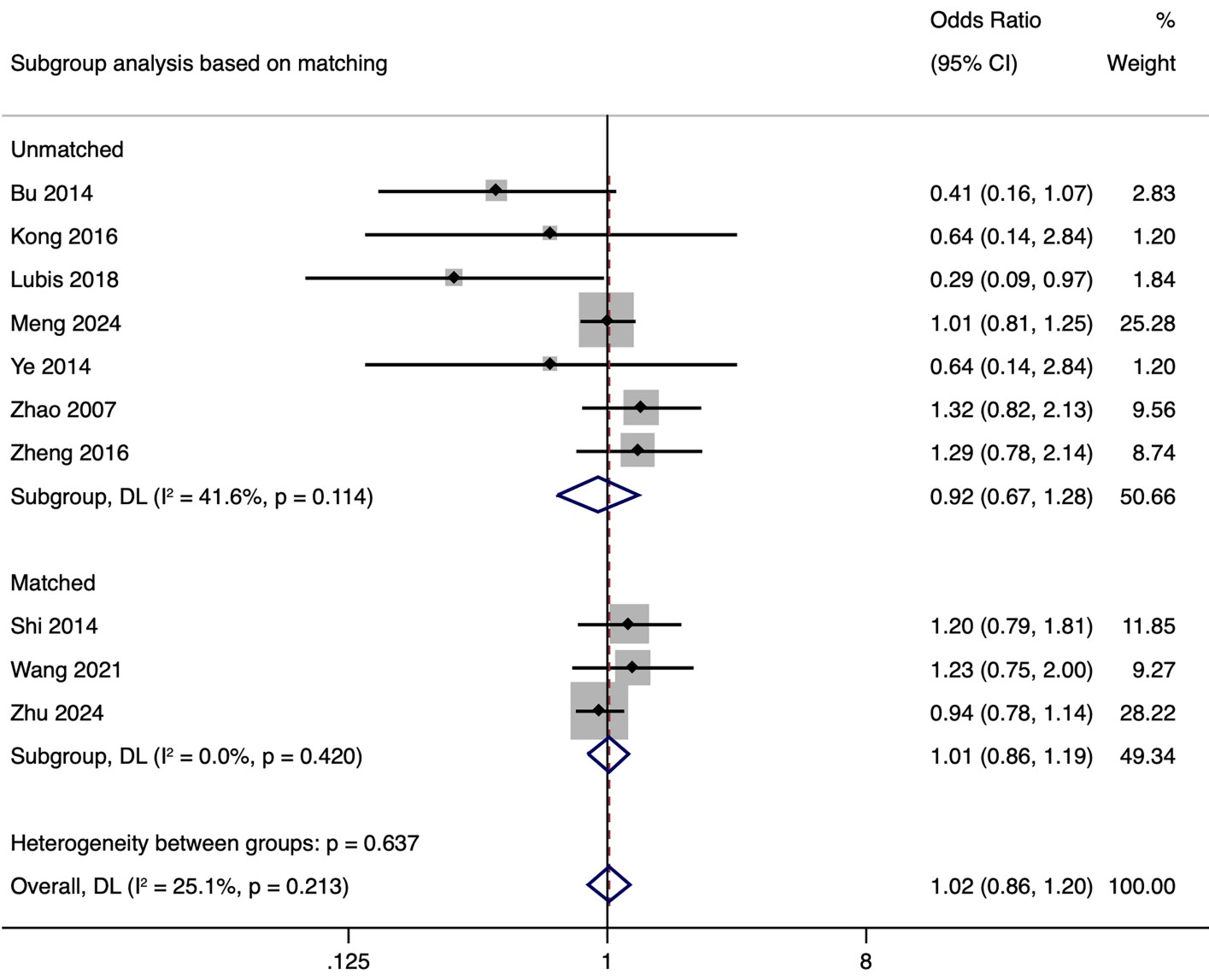

**Figure 3 Subgroup analysis based on matching showing impact of paternal HBV infection on clinical pregnancy rate (per women)** (*Bu et al., 2014; Kong et al., 2017; Lubis et al., 2018; Meng et al., 2024; Shi et al., 2014; Wang et al., 2021; Ye et al., 2014; Zhao et al., 2007; Zheng et al., 2016; Zhu et al., 2024*).

## Impact of paternal HBV infection on live birth rate (per cycle)

The meta-analysis investigating the impact of paternal HBV infection on LBR per cycle included seven studies with 26,324 participants. Pooled OR for LBR among cycles involving HBV-positive men was 0.999, with 95% CI of [0.851 to 1.172] ($p = 0.987$), indicating no statistically significant difference in LBR between cycles with HBV-positive and HBV-negative men (Fig. 8). Heterogeneity among the studies was moderate, with an $I^2$ value of 31.8% and Cochrane Q test $p$-value of 0.185. Sensitivity analysis by excluding

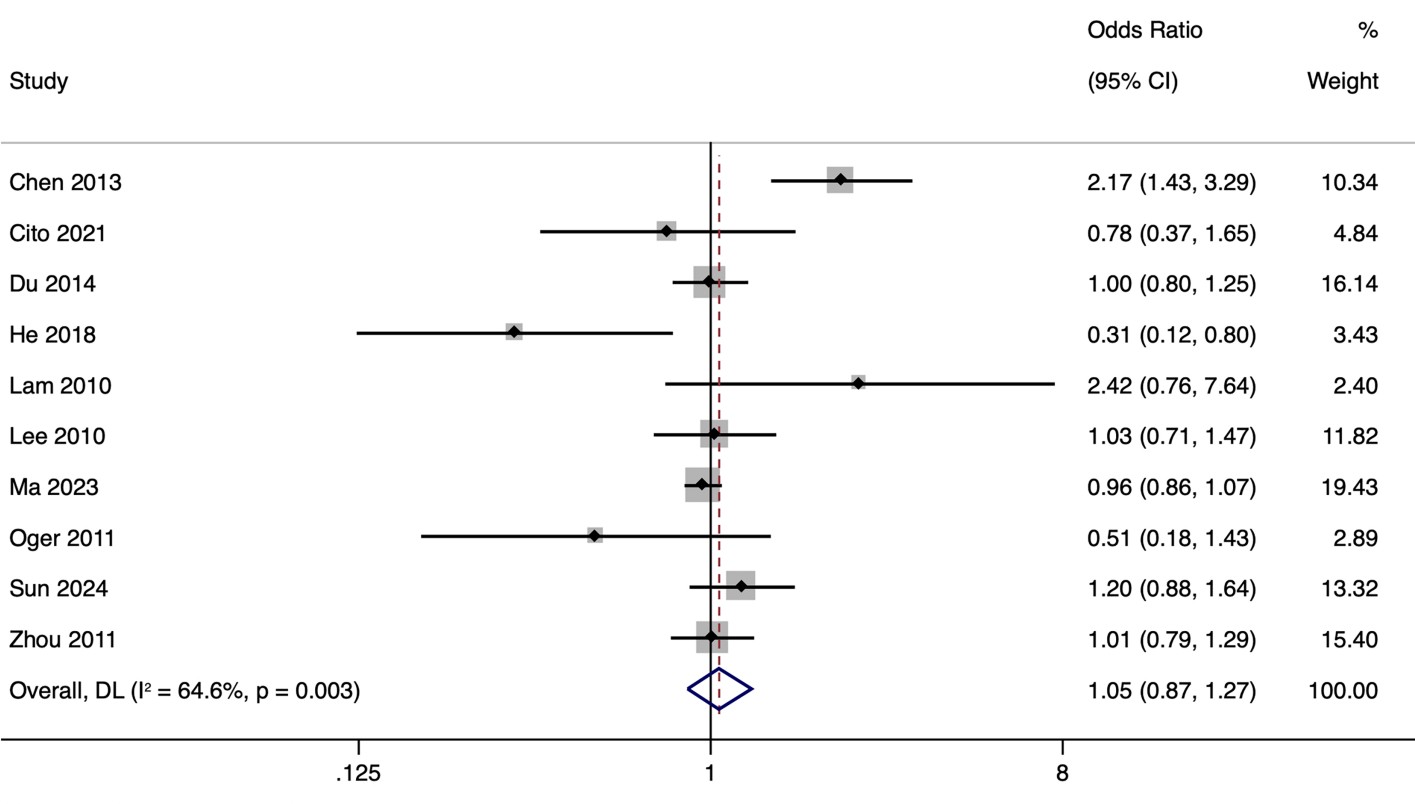

|  | Odds Ratio | % |
| --- | --- | --- |
| Study | (95% CI) | Weight |
| Chen 2013 | 2.17 (1.43, 3.29) | 10.34 |
| Cito 2021 | 0.78 (0.37, 1.65) | 4.84 |
| Du 2014 | 1.00 (0.80, 1.25) | 16.14 |
| He 2018 | 0.31 (0.12, 0.80) | 3.43 |
| Lam 2010 | 2.42 (0.76, 7.64) | 2.40 |
| Lee 2010 | 1.03 (0.71, 1.47) | 11.82 |
| Ma 2023 | 0.96 (0.86, 1.07) | 19.43 |
| Oger 2011 | 0.51 (0.18, 1.43) | 2.89 |
| Sun 2024 | 1.20 (0.88, 1.64) | 13.32 |
| Zhou 2011 | 1.01 (0.79, 1.29) | 15.40 |
| Overall, DL ($I^2$ = 64.6%, p = 0.003) | 1.05 (0.87, 1.27) | 100.00 |

NOTE: Weights are from random-effects model

**Figure 4 Forest plot showing impact of paternal HBV infection on clinical pregnancy rate (per cycle)** (*Chen et al., 2013*; *Cito et al., 2021*; *Du et al., 2014*; *He et al., 2018*; *Lam et al., 2010*; *Lee et al., 2010*; *Ma et al., 2023*; *Oger et al., 2011*; *Sun et al., 2024*; *Zhou et al., 2011*).

studies with higher risk of bias showed that the pooled OR was 1.007 (95% CI [0.745–1.361]), indicating the robustness of the overall pooled effect size.

Subgroup analysis based on matching showed a pooled OR of 0.988 (95% CI [0.832–1.173], $p$ = 0.891) in unmatched and pooled OR of 1.148 (95% CI [0.580–2.274], $p$ = 0.692) in matched studies (Fig. 9). Heterogeneity was moderate in the unmatched studies ($I^2$ = 49.0%; $p$ = 0.118) and low in the matched studies ($I^2$ = 24.8%; $p$ = 0.265). No significant differences in LBR per cycle were detected between pregnancies with HBV-positive and HBV-negative paternal status in both subgroups. The pooled adjusted RR calculated based on the adjusted effect size was 1.024 (95% CI [0.833–1.214]) (Fig. 10). Publication bias assessment was also not done due to the limited number of studies.

## DISCUSSION

This study showed that paternal HBV infection does not significantly influence CPR or LBR per woman or per cycle. These results remained consistent even after subgroup analyses based on study matching status, underscoring the robustness of the findings.

Similar review done by *Xiong et al. (2022)* pooled 11 retrospective cohort studies and found that paternal HBV infection alone was associated with a significant reduction in both the CPR (adjusted RR 0.54 per cycle) and LBR (cRR 0.52 per cycle). In contrast, our

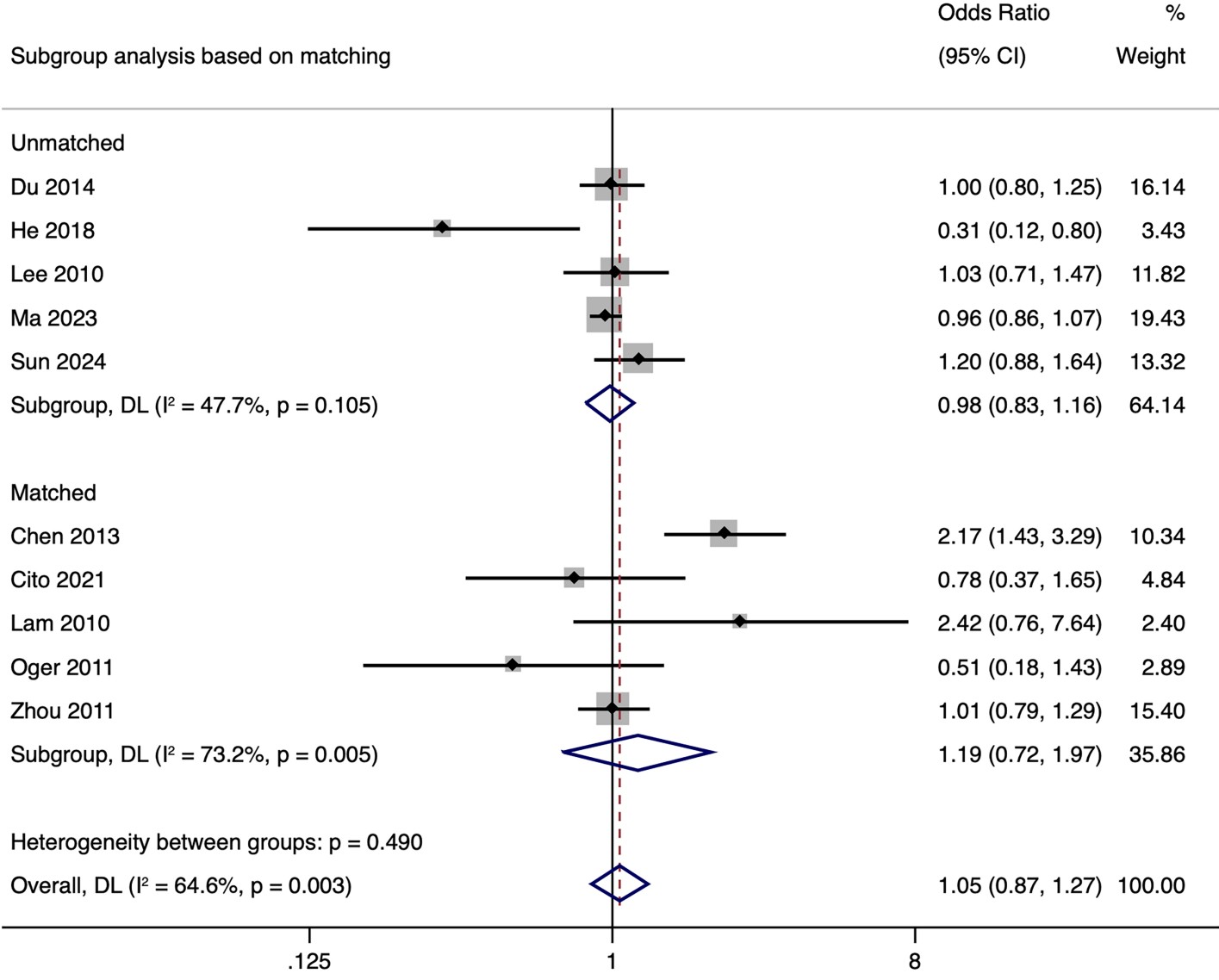

**Figure 5 Subgroup analysis based on matching showing impact of paternal HBV infection on clinical pregnancy rate (per cycle)** (*Chen et al., 2013*; *Cito et al., 2021*; *Du et al., 2014*; *He et al., 2018*; *Lam et al., 2010*; *Lee et al., 2010*; *Ma et al., 2023*; *Oger et al., 2011*; *Sun et al., 2024*; *Zhou et al., 2011*).

meta-analysis which includes 20 studies encompassing over 28,000 ART cycles, have found no significant effect of paternal HBV status on either clinical pregnancy or live birth rates, whether assessed per woman or per cycle. Likewise, *Sun et al. (2024)* in a large single-center IUI cohort that male HBV infection did not adversely impact standard sperm parameters or pregnancy outcomes, although they noted subtle reductions in sperm acrosin activity and altered neonatal outcomes. These apparent discrepancies likely reflect differences in study design (matched *vs* unmatched controls), ART modality (IVF/ICSI *vs* IUI), and the evolution of sperm-washing and viral detection techniques over time. By focusing

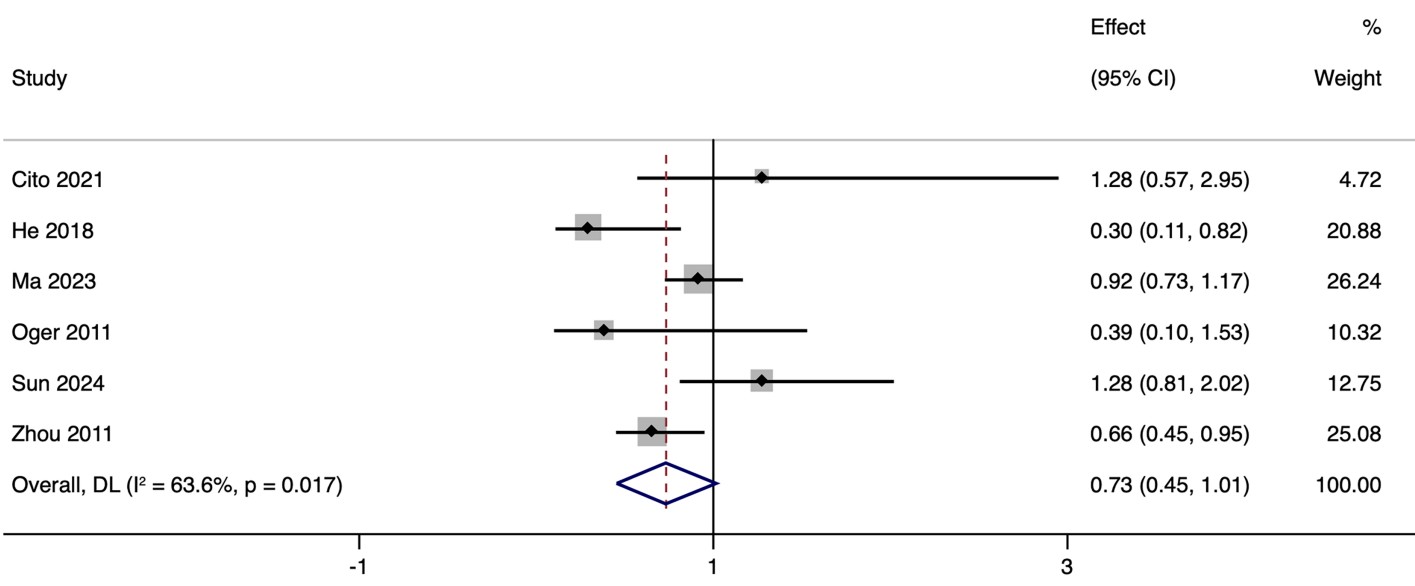

**Figure 6** Forest plot showing impact of paternal HBV infection on clinical pregnancy rate (per cycle) in terms of adjusted estimates (*Cito et al., 2021*; *He et al., 2018*; *Ma et al., 2023*; *Oger et al., 2011*; *Sun et al., 2024*; *Zhou et al., 2011*).

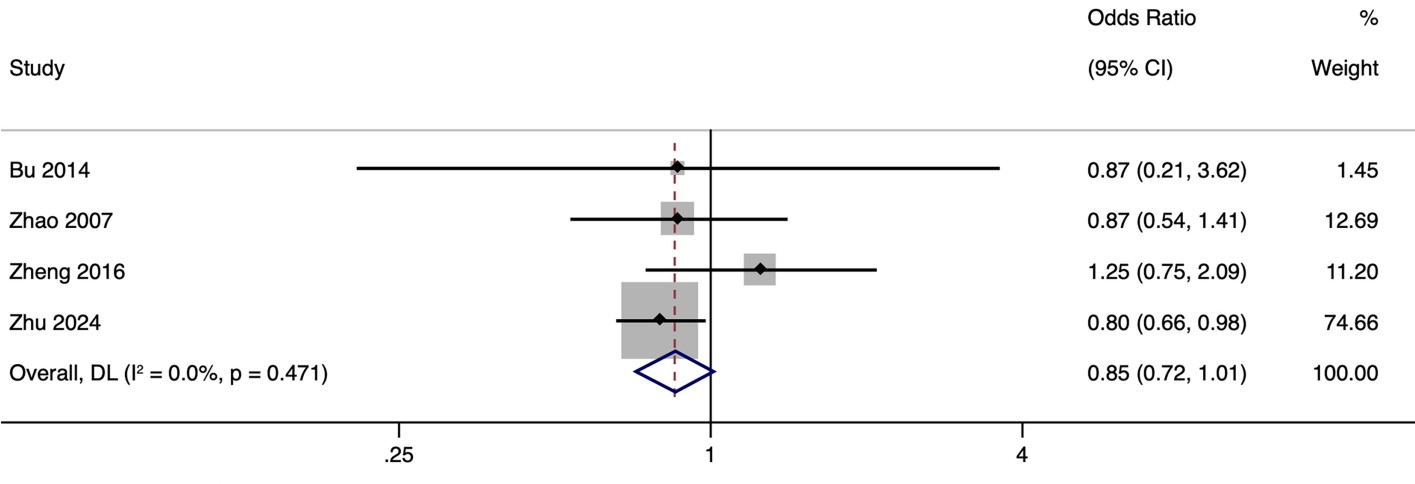

**Figure 7** Forest plot showing impact of paternal HBV infection on live birth rate (per women) (*Bu et al., 2014*; *Zhao et al., 2007*; *Zheng et al., 2016*; *Zhu et al., 2024*).

specifically on IVF/ICSI cycles, stratifying by matching status, and synthesizing a broader temporal span of data, our analysis provides the most rigorous and generalizable evidence to date that paternal HBV infection does not materially compromise ART success.

Although subgroup analyses reduced heterogeneity for some outcomes, we observed that clinical pregnancy rate (per cycle) among matched studies still exhibited high heterogeneity ($I^2 = 73.2\%$, $p = 0.005$). This residual heterogeneity among matched studies for clinical pregnancy rate likely stems from several sources. First, the four matched

**Figure 8 Forest plot showing impact of paternal HBV infection on live birth rate (per cycle)** (*Cito et al., 2021*; *He et al., 2018*; *Lam et al., 2010*; *Ma et al., 2023*; *Meng et al., 2024*; *Oger et al., 2011*; *Sun et al., 2024*).

cohorts differed in their underlying ART protocols—variations in ovarian stimulation regimens (agonist *vs* antagonist-based), embryo grading criteria, and luteal support strategies can all influence clinical pregnancy outcomes. Second, although labelled "matched," not every study adjusted for or matched on the same paternal factors; for example, only two studies reported matching by HBV viral load or measuring semen parameters. Residual confounding by unmeasured paternal characteristics (*e.g.*, variations in HBV viral load, sperm DNA fragmentation, or co-infections) may therefore persist. Third, geographic and ethnic differences (*e.g.*, studies conducted in mainland China *vs* Taiwan) may introduce background variability in baseline fertility rates or genotype-specific HBV effects on spermatogenesis. Fourth, definitions of "clinical pregnancy" were not uniform—some authors considered only the presence of a gestational sac at 6–7 weeks, whereas others required documented fetal cardiac activity; these definitional discrepancies can produce non-comparable effect sizes. Finally, with only four matched studies included, even small between-study differences in population or protocol can generate substantial $I^2$. In light of these factors, future research should focus on prospective, matched or randomized analyses in which (1) paternal HBV viral load and semen quality are uniformly assessed; (2) identical ART protocols and embryo transfer criteria are applied; and (3) a standardized definition of clinical pregnancy (*e.g.*, fetal heartbeat on transvaginal ultrasound at 6 weeks) is used across cohorts to minimize methodological heterogeneity.

Despite observing moderate to substantial heterogeneity across our primary analyses ($I^2 = 64.6\%$ for CPR per cycle and $I^2 = 31.8\%$ for LBR per cycle), sources of this variability

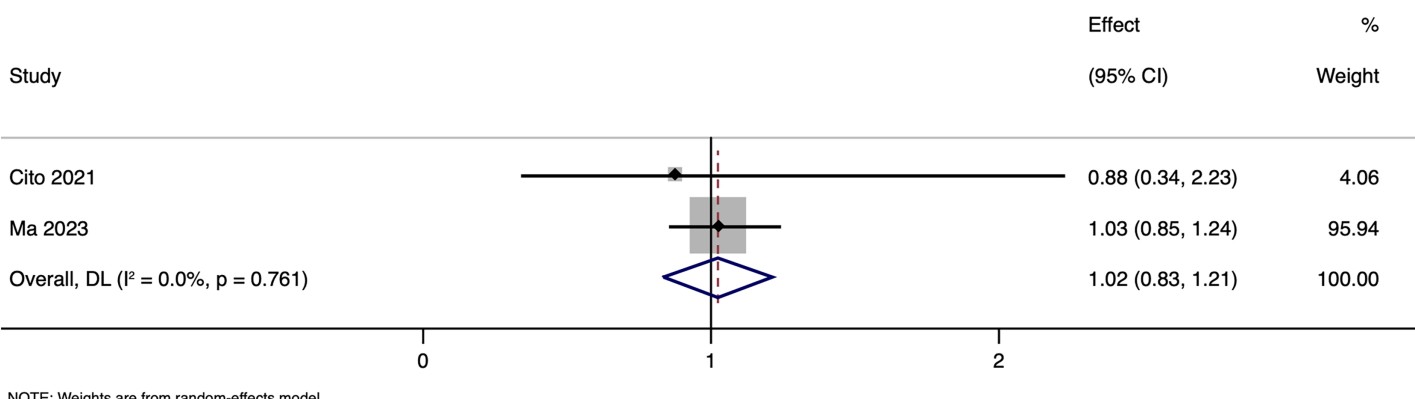

|  | Odds Ratio | % |
|---|---|---|
| Subgroup analysis based on matching | (95% CI) | Weight |

**Unmatched**

| | | |
|---|---|---|
| He 2018 | 0.33 (0.12, 0.95) | 2.20 |
| Ma 2023 | 0.97 (0.87, 1.08) | 44.14 |
| Meng 2024 | 0.98 (0.81, 1.18) | 30.58 |
| Sun 2024 | 1.23 (0.88, 1.73) | 15.95 |
| Subgroup, DL (I² = 49.0%, p = 0.118) | 0.99 (0.83, 1.17) | 92.87 |

**Matched**

| | | |
|---|---|---|
| Cito 2021 | 1.14 (0.49, 2.65) | 3.36 |
| Lam 2010 | 2.32 (0.72, 7.51) | 1.80 |
| Oger 2011 | 0.60 (0.20, 1.85) | 1.98 |
| Subgroup, DL (I² = 24.8%, p = 0.265) | 1.15 (0.58, 2.27) | 7.13 |

Heterogeneity between groups: p = 0.676

| | | |
|---|---|---|
| Overall, DL (I² = 31.8%, p = 0.185) | 1.00 (0.85, 1.17) | 100.00 |

.125   1   8

NOTE: Weights and between-subgroup heterogeneity test are from random-effects model

**Figure 9 Subgroup analysis based on matching showing impact of paternal HBV infection on live birth rate (per cycle) (*Cito et al., 2021*; *He et al., 2018*; *Lam et al., 2010*; *Ma et al., 2023*; *Meng et al., 2024*; *Oger et al., 2011*; *Sun et al., 2024*).**

|  | Effect | % |
|---|---|---|
| Study | (95% CI) | Weight |

| | | |
|---|---|---|
| Cito 2021 | 0.88 (0.34, 2.23) | 4.06 |
| Ma 2023 | 1.03 (0.85, 1.24) | 95.94 |
| Overall, DL (I² = 0.0%, p = 0.761) | 1.02 (0.83, 1.21) | 100.00 |

0   1   2

NOTE: Weights are from random-effects model

**Figure 10 Forest plot showing impact of paternal HBV infection on live birth rate (per cycle) in terms of adjusted estimates (*Cito et al., 2021*; *Ma et al., 2023*).**
have not been fully elucidated. Several factors likely contribute. First, there is considerable clinical heterogeneity in ART protocols, ranging from IVF *vs* ICSI to the use (or not) of advanced sperm-washing techniques, which may differentially mitigate any impact of HBV in semen (*Navabakhsh et al., 2011*; *Jaffe & Brown, 2017*). Second, included cohorts differ in baseline characteristics such as age, infertility etiology, and geographic region (*e.g.*, single-center Chinese studies from 2007–2011 *vs* large multicenter cohorts in 2023–2024), potentially reflecting evolving laboratory standards and patient management practices (*Zhao et al., 2007*; *Ma et al., 2023*). Third, study design features, particularly the presence or absence of matched controls appear influential: matched studies showed minimal heterogeneity ($I^2 = 0\%$ for CPR per woman), whereas unmatched investigations exhibited greater variability, underscoring the role of baseline confounding. Finally, methodological differences in HBV detection (HBsAg *vs* HBV DNA assays), timing of sample collection, and small sample sizes in several reports may amplify random error. Recognizing these sources highlights the need for future investigations with standardized ART and viral-assessment protocols, matched or adequately adjusted designs, and robust sample sizes to reduce heterogeneity and more precisely define the true effect of paternal HBV on ART outcomes.

Several mechanisms may explain the lack of significant impact of paternal HBV infection on ART outcomes. First, while HBV can be present in semen, its ability to affect sperm function and fertilization capacity might be limited (*Han et al., 2021*). The ART process, which includes techniques such as sperm washing and the use of intracytoplasmic sperm injection (ICSI), may mitigate any potential adverse effects of HBV on sperm. Additionally, the immune response of the female reproductive system might neutralize any HBV present in the semen, further reducing the risk of transmission and adverse outcomes (*Zheng et al., 2016*). It is also plausible that the overall health and lifestyle factors of the men undergoing ART might play a more critical role in influencing reproductive outcomes than the presence of HBV alone (*Wang et al., 2021*).

The strengths of our meta-analysis include a comprehensive search strategy, thorough inclusion criteria, and vigorous statistical analysis. The inclusion of both matched and unmatched studies allows for a more nuanced understanding of the potential impact of study design on the outcomes. However, there are also limitations to consider. The heterogeneity among the included studies, particularly in terms of population characteristics and ART protocols, may have influenced the results. Additionally, the limited number of studies available for some outcomes, such as LBR per woman, restricted the ability to conduct more detailed subgroup analyses. Publication bias and the quality of the included studies are concerning, although funnel plots and Egger's test detected no significant publication bias.

Limitations: A small number of studies prevented us from doing subgroup analyses or assessing publication biases for some outcomes. Larger, well-controlled studies should also focus on providing detailed insights into the effect of paternal HBV infection at different stages of ART and on pregnancy outcomes. Future research should explore the potential mechanisms underlying the observed effect. Investigating the potential role of HBV mutations and viral load in influencing reproductive outcomes could also be valuable.

Additionally, studies examining the long-term health outcomes of children born to HBV-positive fathers through ART would provide more comprehensive evidence on the safety and implications of HBV in reproductive health. Research focusing on the psychological and emotional impact of HBV status on couples undergoing ART could also yield valuable insights, helping to tailor supportive interventions that address not only the medical but also the psychosocial aspects of fertility treatment.

Despite these limitations, our findings may provide important implications for clinical practice. They imply that paternal HBV infection should not be a significant concern for couples undergoing ART, and routine screening and management protocols can continue to focus primarily on the maternal HBV status (*Han et al., 2021*; *Wang et al., 2021*). Clinicians can reassure couples that paternal HBV infection is unlikely to impact the success rates of ART procedures, thus encouraging more informed and balanced decision-making. Additionally, healthcare providers should focus on overall lifestyle and health optimization for male partners undergoing ART, as factors such as obesity, smoking, and other infections may play a more significant role in reproductive outcomes than HBV status alone. Comprehensive preconception counseling that includes discussions about lifestyle modifications and general health maintenance could also enhance ART outcomes.

## CONCLUSION

This study provides robust evidence that paternal HBV infection does not significantly affect CPR or LBR in couples going through ART treatments. These findings can help inform clinical practice and alleviate concerns regarding the potential impact of paternal HBV infection. Further studies should confirm our results and explore the mechanisms of the effect in greater detail, ultimately contributing to more effective and evidence-based management of HBV in the context of ART.

### Funding
The authors received no funding for this work.

### Competing Interests
The authors declare that they have no competing interests.

### Author Contributions
- Juanting Gao conceived and designed the experiments, performed the experiments, prepared figures and/or tables, authored or reviewed drafts of the article, and approved the final draft.
- Qiyin Dong performed the experiments, analyzed the data, prepared figures and/or tables, and approved the final draft.
- Liping Shen analyzed the data, prepared figures and/or tables, and approved the final draft.
- Xiuping Zhu performed the experiments, analyzed the data, prepared figures and/or tables, authored or reviewed drafts of the article, and approved the final draft.

## Data Availability

The raw data is available in the Supplemental Files.

## Supplemental Information

Supplemental information for this article can be found online at http://dx.doi.org/10.7717/peerj.19824#supplemental-information.

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
