# Peer review of "Impact of paternal hepatitis B on pregnancy outcomes in couples undergoing assisted reproductive technology treatment: a systematic review and meta-analysis"

_PeerJ, doi:10.7717/peerj.19824_

## Round 0.1 · original submission · Major Revisions

Reviewers identified issues with methodology and reporting.

Reviewer 1 ·

Basic reporting

The manuscript is clear, and the information is well-structured. The abstract effectively summarizes the manuscript's content. The introduction provides relevant details related to the research topic. Additionally, the research gaps are outlined in lines 68–74.

Experimental design

The research method is presented and replicable. The selection criteria for the articles included in the review are explained in lines 107–116. However, five articles (published in 2007, 2010, and 2011) are more than 10 years old. It is recommended to either exclude articles older than 10 years or provide a more detailed justification for their inclusion, particularly regarding why publication year is not considered in the inclusion or exclusion criteria.

Line 133 discusses the risk of bias assessment in article selection. While Table 1 outlines the assessment process, it requires a more detailed explanation of why articles categorized as “low” are still included in the study. It should also be clarified whether the “low” category signifies a high risk of bias, making the article less reliable, or if it indicates otherwise. Providing this clarification will help readers better understand the systematic review process.

Validity of the findings

The presented results demonstrate novelty and successfully achieve the expected research objectives. The discussion effectively interprets the findings and is supported by relevant information. However, lines 248–251 should be reinforced with appropriate references.

Additional comments

The research is engaging and offers a novel contribution to the field of Reproductive Health. However, the article selection process should be clarified to enhance readability and ensure a clear understanding of the systematic review flow. Additionally, the stated research limitations provide valuable guidelines for future studies.

Reviewer 2 ·

Basic reporting

This manuscript is a meta-analysis study to evaluate the association of paternal hepatitis B virus (HBV) infection with clinical pregnancy rate (CPR) and live birth rate (LBR) per woman and cycle in couples with assisted reproductive technology (ART). The authors need to revise the manuscript properly before acceptance. I do believe the manuscript would benefit from language editing.

Experimental design

-

Validity of the findings

-

Additional comments

1. Too much information was provided in the Introduction section, making it difficult to follow. What’s your research question? Moreover, there is a bit too much repetition in the last paragraph of the Introduction section. Enhancing conciseness would improve readability.

2. In general, it would be better to provide detailed definitions of outcomes.

3. I’m wondering if you need to provide a detailed search strategy for the literature search part.

4. P9, Line 133. While the authors indicated the utilization of the Newcastle-Ottawa Scale to assess the risk of bias, I suppose a table including more detailed quality scores in each parameter of the included studies is necessary.

5. P9, Line 143. It wasn’t clear what the difference between matched and unmatched studies was.

6. In addition to I², authors should further use the Cochrane Q test to assess heterogeneity.

7. In Figure 1, it would be better to include more detailed information on why 1,672 records were excluded.

8. I’m wondering if you need to provide an overview of the included studies following the search results.

9. I suppose it would be a good idea to include the number of participants for each group in each study in the forest plots.

10. P10, Line 168. As the authors mentioned in the Results section, supplementary materials should include Funnel plots. However, no Funnel plots were found in the supplementary materials.

11. I’m wondering whether authors could generate a sensitivity analysis after excluding studies that include women with HBV infection. This might make the findings of this study more robust.

12. High heterogeneity was observed in the study, but this was only briefly reported in the results section without exploring the possible causes of heterogeneity in depth. It would be a good idea to refine the content of this article by further exploring the sources of heterogeneity in the discussion.

13. In addition to the match status, it might be better to have more subgroup analyses based on study design and type of ART.

14. In the second paragraph of the Discussion section, could you compare the present results with previous studies more clearly to highlight the differences or new findings?

15. A similar meta-analysis was published in 2022, which only included retrospective cohort studies and observed a significant association between paternal HBV infection and CPR per cycle among couples with ART. Though the authors mentioned that the current review aggregated data from multiple studies with a large population size, it would be a good idea to generate some sensitivity analysis, such as excluding studies that were not published in English or with a high risk of bias, to make the findings more robust.

16. P12, Line 245. Though paternal HBV infection might not influence the pregnancy outcomes among couples with ART, I do not agree that the authors can conclude the findings in this study can reduce unnecessary anxiety and interventions for HBV-positive men seeking to become fathers through ART.

17. It’s a little bit difficult to follow the last three paragraphs in the Discussion section. It would be better to rearrange the sequence.

---

## Round 0.2 · Minor Revisions

Reviewer 1 ·

Basic reporting

no comment

Experimental design

no comment

Validity of the findings

no comment

Reviewer 2 ·

Basic reporting

Thanks for the revised manuscript and responses to my original review. I appreciate the authors’ great effort to address the previous comments. However, I still have a few remaining issues that I would like to raise.

Experimental design

no comment

Validity of the findings

no comment

Additional comments

1. Definition of matched and unmatched study.
As the authors mentioned, “studies as matched if the investigators explicitly paired or frequency‑matched HBV‑positive and HBV‑negative groups on one or more key baseline characteristics” and studies that compared the two groups without such matching or adjustment were considered unmatched.” However, as shown in Table 1, some retrospective cohort studies were defined as matched studies, but others were not. Moreover, one case-control study (Zhao 2007, China) was defined as an unmatched study.
Why do authors think studies with covariate adjustment could be defined as a matched study, but a case-control study could not? Please provide more details to clarify the definition of matched and unmatched studies.

2. High heterogeneity was observed in the analysis of the impact of paternal HBV infection on clinical pregnancy rate (per cycle), which was still observed in matched studies (I2 = 73.2%, P = 0.005) in subgroup analysis. However, the authors didn’t discuss this part further.

3. The x-axis of forest plots and funnel plots is unevenly distributed. Moreover, what’s the effect size in funnel plots?

---

## Round 0.3 · accepted · Accept

The revised version has adequately addressed reviewers concerns.

Reviewer 2 ·

Basic reporting

Thanks for the revised manuscript and responses to my reviews.
This version of the manuscript is substantially improved.

Experimental design

-

Validity of the findings

-